Tip dating supports novel resolutions
of controversial relationships among
early mammals. *Proc. R. Soc. B* **287**: 20200943.

evolution, palaeontology

Bayesian, Haramiyida, Allotheria, *Juramaia*,
mammals, mammaliaforms

**Author for correspondence:**
Benedict King
e-mail: benking315@gmail.com

Electronic supplementary material is available
online at https://doi.org/10.6084/m9.figshare.
c.4994606.

# Tip dating supports novel resolutions of controversial relationships among early mammals

Benedict King[1,2] and Robin M. D. Beck[3]

[1]Naturalis Biodiversity Center, Leiden, the Netherlands
[2]College of Science and Engineering, Flinders University, Adelaide, South Australia, Australia
[3]School of Environmental and Life Sciences, University of Salford, Salford M5 4WT, UK

BK, 0000-0002-9489-8274; RMDB, 0000-0002-7050-7072

The estimation of the timing of major divergences in early mammal evolution is challenging owing to conflicting interpretations of key fossil taxa. One contentious group is Haramiyida, the earliest members of which are from the Late Triassic. Many phylogenetic analyses have placed haramiyidans in a clade with multituberculates within crown Mammalia, thus extending the minimum divergence date for the crown group deep into the Triassic. A second taxon of interest is the eutherian *Juramaia* from the Middle–Late Jurassic Yanliao Biota, which is morphologically very similar to eutherians from the Early Cretaceous Jehol Biota and implies a very early origin for therian mammals. Here, we apply Bayesian tip-dated phylogenetic methods to investigate these issues. Tip dating firmly rejects a monophyletic Allotheria (multituberculates and haramiyidans), which are split into three separate clades, a result not found in any previous analysis. Most notably, the Late Triassic *Haramiyavia* and *Thomasia* are separate from the Middle Jurassic euharamiyidans. We also test whether the Middle–Late Jurassic age of *Juramaia* is 'expected' given its known morphology by assigning an age prior without hard bounds. Strikingly, this analysis supports an Early Cretaceous age for *Juramaia*, but similar analyses on 12 other mammaliaforms from the Yanliao Biota return the correct, Jurassic age. Our results show that analyses incorporating stratigraphic data can produce results very different from other methods. Early mammal evolution may have involved multiple instances of convergent morphological evolution (e.g. in the dentition), and tip dating may be a method uniquely suitable to recognizing this owing to the incorporation of stratigraphic data. Our results also confirm that *Juramaia* is anomalous in exhibiting a much more derived morphology than expected given its age, which in turn implies very high rates of evolution at the base of therian mammals.

## 1. Introduction

Allotherians are an extinct group of mammaliaforms, primarily known from the Mesozoic, that are currently the subject of conflicting phylogenetic hypotheses (figure 1). Allotherians share a number of dental apomorphies, most notably postcanines with multiple cusps in longitudinal rows (superficially resembling those of some therian mammals, such as rodents), and they include haramiyidans, multituberculates and gondwanatherians [1–6]. Some phylogenetic analyses have supported monophyly of Allotheria, within (crown-clade) Mammalia [2,7–10] (figure 1, topology 1). Conversely, others have recovered haramiyidans outside Mammalia, but with multituberculates remaining within Mammalia [3,11,12] (figure 1, topology 2a), suggesting that allotherian dental apomorphies have evolved more than once. Finally, two studies recovered diphyletic haramiyidans, with the euharamiyidans forming a clade with

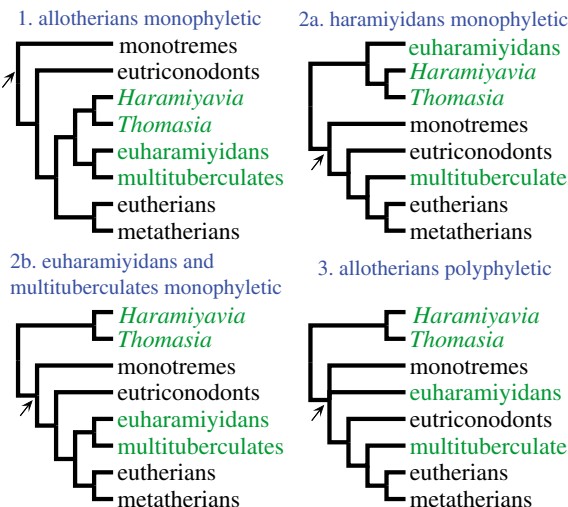

**Figure 1.** Summary of the main hypotheses for the relationships of 'allotherians'. Numbers refer to the number of different independent clades of 'allotherians'. Arrows indicate the mammalian crown node. Three independent clades is a novel result from this study. (Online version in colour.)

multituberculates within crown mammals, but the Triassic species *Haramiyavia* and *Thomasia* falling outside the crown group [13,14] (figure 1, topology 2b).

Monophyly versus polyphyly of Allotheria has major implications for our understanding of Mesozoic mammal evolution, leading to different scenarios for the evolution of numerous dental and skeletal features, including the so-called Definitive Mammalian Middle Ear, in which the angular, articular, prearticular, and quadrate have become entirely auditory in function, and are fully separated from the jaw joint [9,15,16]. It also affects interpretations of the age of Mammalia: if Late Triassic haramiyidans such as *Haramiyavia* and *Thomasia* fall within the crown-clade, then the split between monotremes and therians must be at least this old [8]; if they fall outside the crown-clade, this split could be considerably younger, as it would render *Asfaltomylos* and *Henosferus* (which appear to be early relatives of monotremes) from the Middle Jurassic of Patagonia the oldest known members of the crown-clade [17].

Another fossil mammal that has been the subject of recent discussion is the eutherian *Juramaia sinensis* from the Middle–Late Jurassic (164–159 Ma) Linglongta Biota (the younger of the two phases composing the Yanliao/Daohugou Biota) from the Lanqi/Tiaojishan Formation of China [18,19]. Based on its known morphology, *Juramaia* has been argued by some authors [20–22] to be 'unexpectedly advanced' for its age, as it closely resembles eutherians from the much younger (ca 126 Ma) Jehol Biota [20,22]. By contrast, the same has not been argued for other mammaliaforms from the Yanliao Biota. However, whether or not the known morphology of *Juramaia* is 'unexpected' given its age has never, to our knowledge, been quantitatively tested.

Tip-dated phylogenetic methods [23], which include morphological and stratigraphic data in a single analytical framework, are a promising avenue to investigate these issues. The wide time difference between the earliest known haramiyidans (Late Triassic) and the oldest known multituberculates (Middle Jurassic) [5,24] suggests that their similarities may be the result of convergent evolution, and incorporating stratigraphic data into phylogenetic analysis means that this temporal disparity is taken into account [25]. Another use of

tip dating is to use the morphological data to inform the ages of fossils with uncertain dates [26,27]. Given that the known morphology of *Juramaia* has been identified as 'unexpectedly advanced' [20–22], it can be used to test whether tip dating continues to support a Middle–Late Jurassic age when its age is allowed to vary. Here, we apply tip dating to recent datasets of Mesozoic mammals to investigate the relationships of the haramiyidans, and to test the congruence between the known morphology and age of *Juramaia* and other Yanliao mammaliaforms.

## 2. Material and methods

Our focal dataset was taken from Huttenlocker *et al.* [3], which comprises 538 morphological characters scored for 125 mammaliaforms and non-mammaliaform cynodonts. Because the sampling of Cenozoic taxa in this dataset was extremely sparse relative to Mesozoic taxa, extant and Cenozoic fossil taxa were pruned from the dataset, and invariant characters in this reduced dataset were deleted, leaving 96 taxa and 507 characters. Tip-dated Bayesian analyses were performed in BEAST v. 2.5.2 [28]. The Markov model for variable characters (hereafter Mkv) was used [29], with a gamma distribution (with four rate categories) to account for rate variation across sites. Characters were partitioned according to the number of character states. The clock model was an uncorrelated lognormal clock [30], and the tree prior was a sampled-ancestor fossilised birth–death model [31]. Tip dates were assigned uniform priors across the range of uncertainty for each taxon. The analysis was run for 1 billion generations, sampling every 500 000. Convergence of four independent runs was confirmed in TRACER [32], and the R package RWTY [33]. To investigate conflicts between the different parts of the dataset of Huttenlocker *et al.* [3], and further test allotherian relationships, the following character subsets were analysed individually: craniodental, dental only, and postcranial only. Undated Bayesian analyses were performed in MRBAYES [34], again using the Mkv model with a gamma distribution (with four rate categories) to account for rate variation across sites. Four independent runs, each with four chains, were run for 10 million generations, sampling every 5000. Parsimony analyses in TNT [35] employed new technology search, using sectorial search and tree fusing with default settings for 1000 random addition sequences, followed by tree bisection and reconnection swapping to fully explore tree islands. We also ran a constrained parsimony analysis with a negative constraint on haramiyidan monophyly.

To further test the extent to which tip dating could overturn topologies supported under other methods, similar tip-dated analyses were run on the datasets of Krause *et al.* [2] and Wang *et al.* [15], both of which originally recovered a monophyletic Allotheria. Extant taxa were pruned, as above, resulting in datasets of 81 taxa, 448 characters and 89 taxa, 473 characters, respectively. Tip-dated analysis of the Krause *et al.* [2] dataset showed very poor mixing (caused by alternative likelihood peaks representing monophyly or polyphyly of Allotheria) and was therefore run for 32 independent runs, each of a billion generations, to obtain reliable estimates of the relative posterior probabilities of the two phylogenetic hypotheses. Results from each run were thinned (sampling every 5 million generations) and, following removal of a 50% burn-in from each run, combined for further analysis.

We also ran an analysis of the Huttenlocker *et al.* [3] dataset with a wider prior age range for *Juramaia*. This represents a quantitative test of the ability of tip dating to infer the age of *Juramaia* based on its known morphology. The tip age prior for *Juramaia* was modified to a Laplace distribution centred on 161 Ma, with a scale parameter of 8. This represents a strong prior expectation

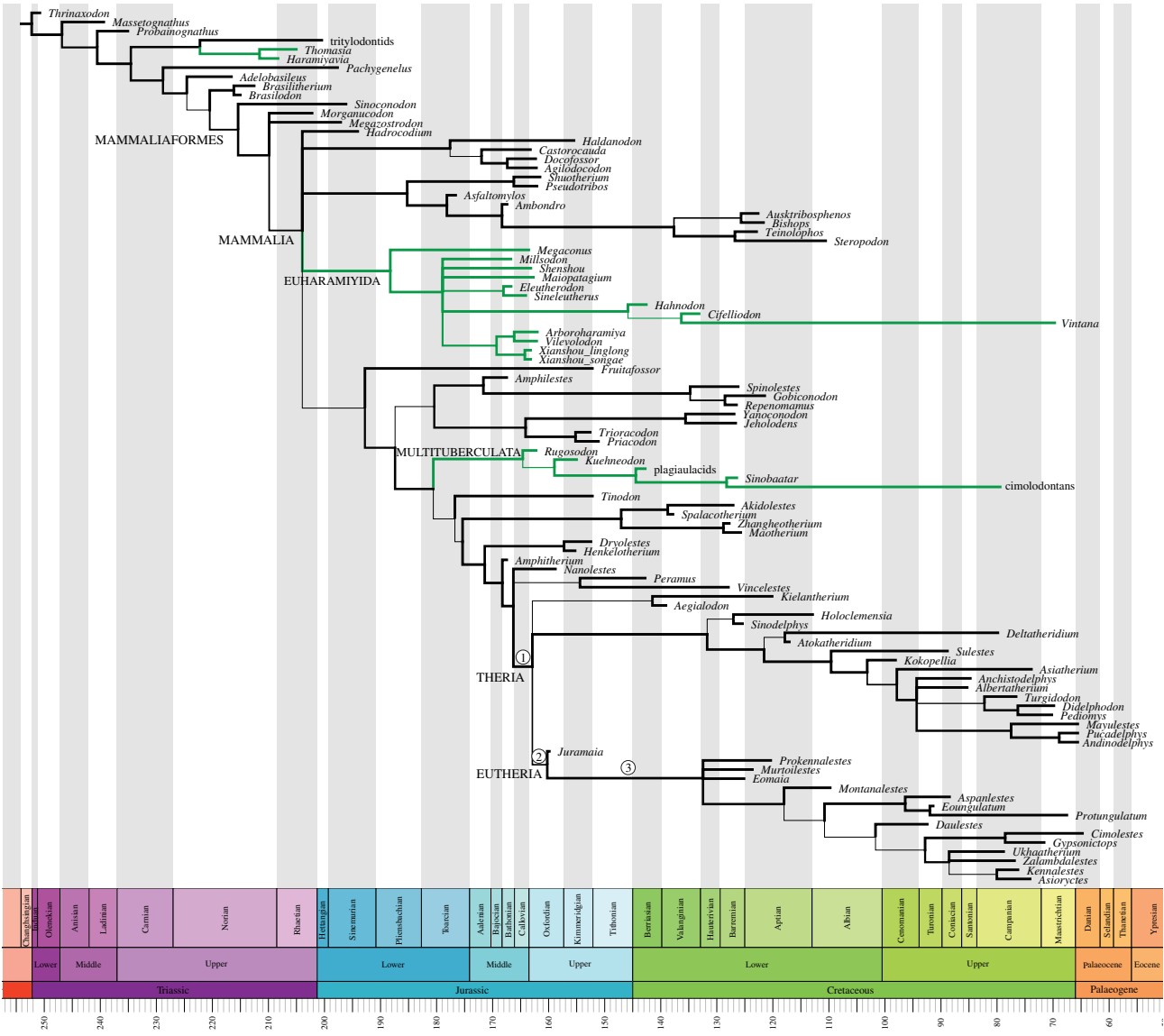

**Figure 2.** Fifty per cent majority rule consensus tree for Mesozoic mammals from a tip-dated analysis of the dataset in Huttenlocker *et al.* [3]. Allotherians (*Haramiyavia*, *Thomasia*, euharamiyidans, multituberculates, hahnodontids and the gondwanatherian *Vintana*) are in green. Branch widths are proportional to posterior probability (between 0.5 and 1.0). Labelled branches (1–3) indicate the branches used for the rate analysis in the electronic supplementary material, figure S13. (Online version in colour.)

that *Juramaia* is Jurassic in age (with 90% of the prior probability density between 143 and 179 Ma), but owing to the absence of hard maximum or minimum bounds, dates outside this range are permitted. The other taxa from the Yanliao Biota in this dataset (*Agilodocodon*, *Arboroharamiya*, *Castorocauda*, *Docofossor*, *Maiopatagium*, *Megaconus*, *Pseudotribos*, *Rugosodon*, *Shenshou*, *Vilevolodon*, *Xianshou linglong* and *Xianshou songae*), were given the same Laplace distribution prior in separate analyses, to test the effectiveness of this method. Extraction of branch rates from the consensus trees for plotting (electronic supplementary material, figure S13) used the R package OutbreakTools [36].

## 3. Results

### (a) Allotherian relationships

Tip-dated analysis of our focal dataset, modified from Huttenlocker *et al.* [3], resulted in allotherian taxa falling into three separate clades (figure 2). The Late Triassic haramiyidans *Haramiyavia* and *Thomasia* are placed outside Mammaliaformes, in a strongly supported clade with tritylodontids (posterior probability (PP) = 0.91). The Middle Jurassic euharamiyidans, Early

Cretaceous hahnodontids, and the Late Cretaceous Madagascan gondwanatherian *Vintana*, by contrast, collectively form a strongly supported clade (PP = 1.00) within Mammaliaformes, although our phylogeny is insufficiently well resolved to indicate whether or not this is within crown-clade Mammalia. Finally, the multituberculates form a third strongly supported clade (PP = 1.00), within Mammalia.

Both undated Bayesian and parsimony analysis recovered monophyletic Haramiyida (table 1). Parsimony analysis with a negative constraint on haramiyidan monophyly (i.e. preventing *Haramiyavia* and *Thomasia* from forming a clade with euharamiyidans) produce trees that are only two steps longer (representing just a 0.1% increase in tree length) than the unconstrained trees. Constrained and unconstrained trees were not significantly different ($p = 0.87$) under the Templeton test [37].

Support for monophyly of Allotheria and of Haramiyida is driven by dental characters, and it should be noted that *Thomasia* is known only from isolated teeth and that *Haramiyavia* is also represented almost exclusively by dental characters. Analysis of craniodental or dental only character subsets led to allotherians falling into progressively fewer

**Table 1.** Support for different configurations of the 'Allotheria' across phylogenetic reconstruction methods and data subsets. (Topologies refer to the number of independent clades formed by the three allotherian groups (figure 1): numbers are posterior probabilities in percentage form. Shaded cells refer to the topology found in the consensus tree (50% majority rule for Bayesian and strict consensus for parsimony).)

| method | tip-dated Bayesian | | | | undated Bayesian | | | | parsimony | | | |
|---|---|---|---|---|---|---|---|---|---|---|---|---|
| topology | 1 | 2a | 2b | 3 | 1 | 2a | 2b | 3 | 1 | 2a | 2b | 3 |
| complete dataset | 0.0 | 5.7 | 0.0 | 94.3 | 0.0 | 71.9 | 0.0 | 28.0 | | | | |
| craniodental | 0.2 | 59.5 | 12.9 | 27.4 | 0.3 | 1.7 | 92.8 | 5.0 | | | | |
| dental | 96.8 | 0.0 | 0.0 | 0.0 | 98.9 | 0.0 | 0.0 | 0.0 | | | | |

separate clades across tip-dated, undated and parsimony methods (table 1; electronic supplementary material, figures S3–S5). Strong support for allotherian polyphyly (i.e. three independent clades) is only found under tip dating on the full dataset, whereas all methods support allotherian monophyly when dental characters are considered in isolation. Tip-dated analysis of postcranial characters only also recovers separate euharamiyidan and multituberculate clades (electronic supplementary material, figure S5), but *Haramiyavia*, *Thomasia*, hahnodontids and *Vintana* could not be included in this analysis as postcranial remains have not been described for them [13,38].

Tip dating using the Wang *et al.* [15] dataset recovered a diphyletic Haramiyida (electronic supplementary material, figure S6), with euharamiyids and multituberculates forming a clade distant from *Haramiyavia + Thomasia* (figure 1, topology 2b). The dataset of Krause *et al.* [2] led to a more complex result, as the sample of post-burn-in trees includes some topologies in which Allotheria is polyphyletic and others in which it is monophyletic. This analysis showed 'twin peak' behaviour of the prior and likelihood traces (electronic supplementary material, figure S7). These peaks correspond to the two different tree topologies regarding Allotheria. One peak, where the Late Triassic *Haramiyavia* and *Thomasia* formed a clade with other allotherians (essentially the parsimony result) had a low prior (or tree model likelihood) but a high likelihood (electronic supplementary material, figure S8–S9). The other peak, which had *Haramiyavia* and *Thomasia* closer to the root of the tree, and separated from other allotherians, had a higher prior and lower likelihood (electronic supplementary material, figure S10). Overall, allotherian monophyly remained the preferred hypothesis, found in 73% of the posterior sample, compared with 27% showing polyphyly of Allotheria. *Arboroharamiyavia*, the only euharamiyidan included in the Krause *et al.* [2] dataset, was always recovered with multituberculates. A constrained parsimony search revealed that polyphyly of Allotheria requires four additional steps (a 2.23% increase in tree length) compared to the unconstrained analysis (which recovers allotherian monophyly). However, constrained and unconstrained trees were not significantly different ($p = 0.68$) under the Templeton test [37].

## (b) Age of *Juramaia*

Rerunning the analysis on the Huttenlocker *et al.* [3] dataset without a hard upper or lower bound on the age of *Juramaia* had no effect on the recovered relationships of haramiyidans and multituberculates: haramiyidan diphyly (and allotherian

triphyly) was still recovered (electronic supplementary material, figure S11). Strikingly, however, this analysis revealed a strong signal in the data supporting a post-Jurassic age for *Juramaia* (figure 3). The mean estimated age for *Juramaia* was 123.5 Ma, almost exactly the same as the age of the Jehol Biota, from where several fossil eutherians are known that are morphologically similar to *Juramaia* [22]. The 95% highest posterior density (HPD) interval was 106.3–137.6 Ma, entirely within the Early Cretaceous. This contrasts with the results from the other Yanliao Biota mammaliaforms. When these were assigned the same Laplace distribution age prior as *Juramaia*, the resulting age estimates were always Jurassic. *Megaconus* resulted in the most inaccurate age estimate (mean 173.6 Ma), but the 95% HPD interval (154.8–194.8 Ma) comfortably overlapped the true age of the Yanliao Biota. For all other taxa, mean age estimates were between 156.8 Ma (*Rugosodon*) and 164.1 Ma (*Castorocauda*) and 95% HPD intervals fell between 142.6 Ma (lower bound for *Rugosodon*) and 182.4 Ma (upper bound for *Maiopatagium*). The *Juramaia* result may be partly driven by low sampling of eutherians during the Early Cretaceous (electronic supplementary material, text; figure S12): estimating the age of *Rugosodon* after deleting the similarly aged multituberculate *Kuehneodon* and plagiaulacids resulted in a wide age estimate (95% HPD 114.4–164.5 Ma).

The age of *Juramaia* also has a significant effect on estimated rates of evolution (electronic supplementary material, figure S13a). When *Juramaia* is assigned its correct, Middle–Late Jurassic age, rates of evolution on the branch leading to crown Theria, and the branch leading to Eutheria, are estimated to be the highest across the entire tree and nearly 10 times higher than the average for all branches, as previously reported by Close *et al.* [39]. The rate on the branch leading to Eutheria excluding *Juramaia* is however very low, suggesting a 50-fold decrease in evolutionary rates in eutherians across the Jurassic–Cretaceous boundary. However, when the age of *Juramaia* is allowed to vary (resulting in the estimation of an Early Cretaceous age), rates of evolution on these three branches are far more similar, resulting in approximately constant rates during early eutherian evolution (electronic supplementary material, figure S13b).

## 4. Discussion

### (a) Allotherian relationships

The results of our tip-dated analysis of the Huttenlocker *et al.* [3] dataset suggest that the dental similarities proposed to unite Allotheria are homoplastic, and that they evolved at

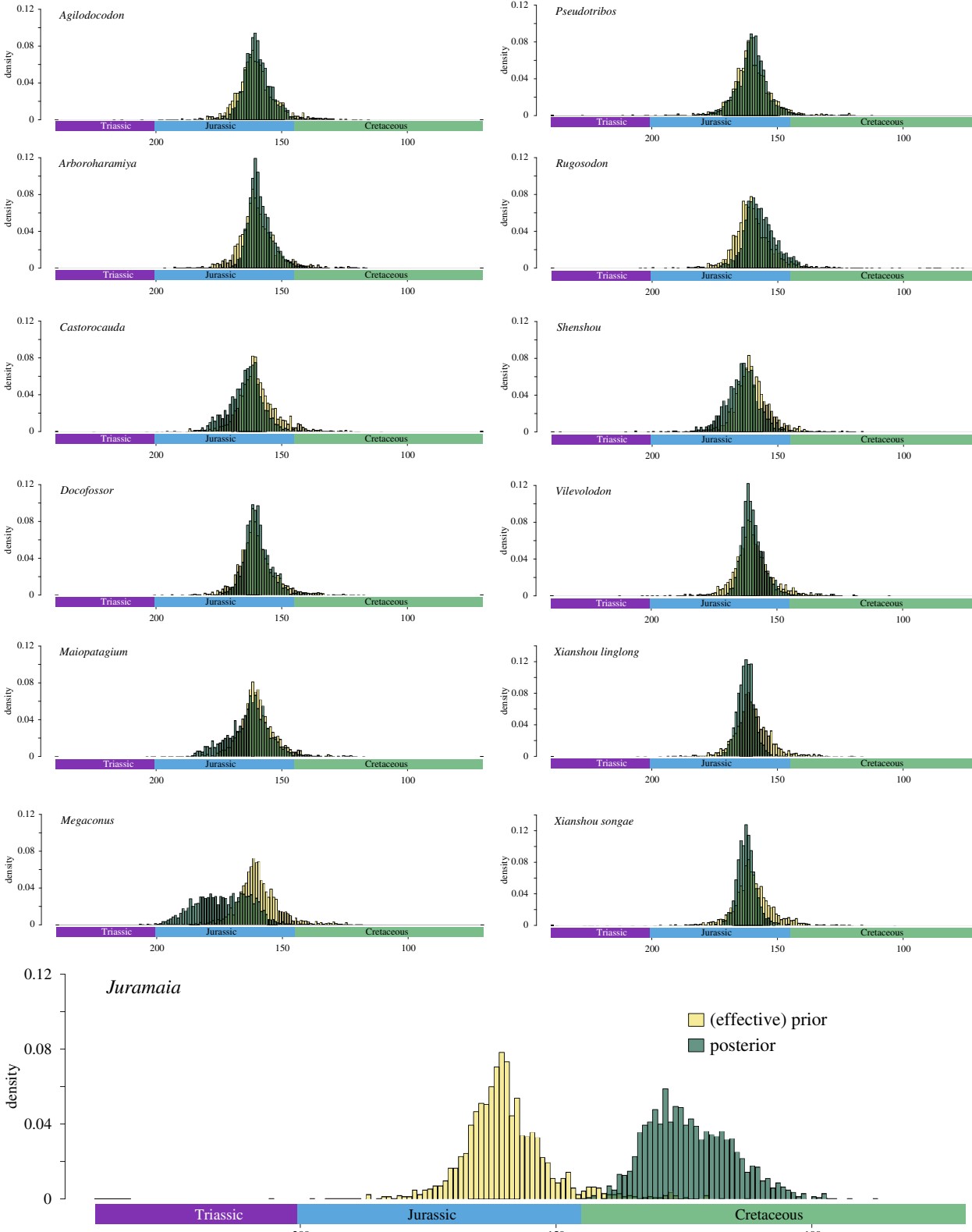

**Figure 3.** The morphological data have a strong signal towards an Early Cretaceous rather than a Jurassic age for *Juramaia*. Age estimates for members of the Yanliao Biota, analysed with a Laplace distribution prior centred on 161 Ma, therefore representing a conservative test. Data for *Juramaia* strongly contradicts the prior, in contrast to the other Yanliao mammaliaforms, all of which retain a Jurassic age. (Online version in colour.)

least three times independently: once in the common ancestor of *Haramiyavia + Thomasia* and tritylodontids, once in the common ancestor of euharamiyidans, hahnodontids and gondwanatherians, and once in multituberculates (*contra* [4,5,8,15,40]). Notably, a recent study found that dental characters in mammals are more prone to homoplasy than characters from the rest of the skeleton [41]. Our results are congruent with recently discovered morphological

differences between Triassic haramiyidans and the euharamiyidans. In particular, *Haramiyavia* retains a prominent postdentary trough [13], a plesiomorphic feature indicating that it lacked fully detached ear ossicles, whereas in most euharamiyidans (with the notable exceptions of *Megaconus* and *Vilevolodon* [12,16,42]) this trough is either very small or absent [7–10,16]. In some ways, our results represent a compromise between differing views on whether

haramiyidans are crown- or stem-mammals: euharamiyidans fall within or near the crown-clade, whereas *Haramiyavia* + *Thomasia* fall outside. Our analysis places *Haramiyavia* and *Thomasia* in a clade with tritylodontids, a result that may be the result of insufficient sampling of non-mammaliaform cynodont characters and taxa, and which we consider in need of further testing (see detailed discussion in the electronic supplementary material).

The recovered phylogenetic relationships of allotherians depend on both the dataset and the method used. Tip-dated methods invariably push the results towards splitting up the allotherians, but the extent of this depends on the data matrix. For the Krause et al. [2] and Wang et al. [15] datasets, which originally recovered allotherian monophyly (figure 1, topology 1), tip dating leads to increased support for two independent lineages (figure 1, topology 2b), a topology possibly supported by recently discovered morphological similarities between early multituberculates and euharamiyids [24]. For the dataset from Huttenlocker et al. [3], which originally recovered separate haramiyidans and multituberculates (topology 2a), tip dating decisively supports three independent lineages (topology 3).

The relative influence of stratigraphic and morphological data in tip-dated analyses remains an underexplored issue. Tip dating of the Huttenlocker et al. [3] dataset results in strong support for polyphyly of Allotheria, including diphyly of the haramiyidans, a result that requires only two additional steps under parsimony. By contrast, the dataset of Krause et al. [2] has stronger morphological support for allotherian monophyly. Analysis of this dataset flips between allotherian polyphyly and monophyly, and allotherian polyphyly requires four additional parsimony steps over monophyly. In the case of the Krause et al. [2] dataset, the stronger morphological signal for allotherian monophyly is therefore not fully overruled by the stratigraphic evidence. These results suggest that the stratigraphic data only become influential on tree topology when morphological support for conflicting topologies is weak. The effect of stratigraphic age on haramiyidan relationships is analysed quantitatively in the electronic supplementary material.

## (b) Age of *Juramaia*

For some datasets at least, Bayesian tip dating appears to perform relatively well at estimating the ages of tips when treated as unknown [26], although 95% HPDs can be wide [43]. However, in this case, this method failed to accurately identify *Juramaia* as Middle–Late Jurassic in age, confirming that this taxon is characterised by a morphology that is unusually derived given its age. The Jurassic age of *Juramaia* suggests unusually rapid rates of evolution at the base of therians and eutherians, followed by a 50-fold rate decrease and a period of exceptionally slow eutherian morphological evolution during the Early Cretaceous [39].

The *Juramaia* result requires further scrutiny owing to low sampling and phylogenetic uncertainty of early therian mammals (electronic supplementary material, text; figure S12). Our result is largely driven by two taxa, both of which are known from single specimens: *Juramaia* and *Eomaia*. The highly incomplete record of early eutherians [22] makes it difficult to reach robust conclusions regarding the macroevolution of the group, and these may change with future discoveries. *Juramaia* has also been considered to be a stem therian by some authors [44], a phylogenetic position that would be more consistent with its age. Finally, *Sinodelphys* has recently been proposed to be a eutherian rather than a metatherian [22]. If this is the case, it could alter branch length estimates, and influence inferred patterns of early eutherian evolution.

Data accessibility. Full data, analysis code, files and results are available as part of the electronic supplementary material.

Authors' contributions. B.K. and R.M.D.B. designed the study and wrote the paper. B.K. performed analyses and produced figures.

Competing interests. We declare we have no competing interests.

Funding. We have received no funding for this article.

Acknowledgements. B.K. thanks Martin Rücklin for generously supporting the completion of this paper. We thank Brian Davis, Guillermo Rougier, Erik Seiffert, Ian Corfe, Mike Lee, David Grossnickle, Lucas Weaver and an anonymous reviewer for their insightful and constructive comments that much improved the paper. TNT is made freely available through the Willi Hennig society.

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
