## [Reviewer comments · Proceedings of the Royal Society B: Biological Sciences]

Review History

RSPB-2020-0055.R0 (Original submission)

Review form: Reviewer 1 (Thomas Halliday)

Recommendation

Accept with minor revision (please list in comments)

Scientific importance: Is the manuscript an original and important contribution to its field?

Good

General interest: Is the paper of sufficient general interest?

Acceptable

Quality of the paper: Is the overall quality of the paper suitable?

Good

Is the length of the paper justified?

Yes

Should the paper be seen by a specialist statistical reviewer?

No

Do you have any concerns about statistical analyses in this paper? If so, please specify them explicitly in your report.

No

It is a condition of publication that authors make their supporting data, code and materials available - either as supplementary material or hosted in an external repository. Please rate, if applicable, the supporting data on the following criteria.

Is it accessible?

Yes

Is it clear?

Yes

Is it adequate?

No

Do you have any ethical concerns with this paper?

No

Comments to the Author

King and Beck use up-to-date tip-dating analyses to investigate the relationships among Mesozoic mammals, particularly early eutherians and haramiyids. They conclude that tip-dating alters evolutionary interpretations from other phylogenetic methods by explicitly including stratigraphic data, and note that Juramaia is more similar to later forms than would be expected given its age.

The paper is to my mind strong; my comments below are mostly to do with the presentation of the Juramaia issue and a missing piece of the methodological puzzle.

'UNEXPECTED' MORPHOLOGY OF JURAMAIA

The word "unexpected" comes across as potentially problematic from a philosophical perspective. The basic argument in favour of it would be "Juramaia looks like Cretaceous mammals, but is actually Jurassic. That's unexpected". But what you can and do quantify are branch lengths and evolutionary rate, and you can test for expectation only really on those parameters. In both of the citations given for Juramaia being "unexpectedly advanced", that phrase is only used exactly for Durlstotherium etc. rather than the Juramaia -Eutheria sister relationship by Bi et al., although the sentiments are certainly similar.

However, both the Bi et al. and Meng papers discuss two possible explanations for the similarity between Juramaia and Eomaia. These are (a) an incorrect age estimate for Juramaia or (b) early appearance of eutherian-type dentition and then exceptionally slow rates of evolution. This is the justification for qualitatively noting that something is "unexpected". They also both caution that the fact that Juramaia is represented by a single specimen makes identification of a pattern practically impossible.

I think it's worth noting that the age of the Yanliao Biota is now pretty solidly fixed in the Middle-Late Jurassic - a single adjective when bringing up the biota for the first time would suffice - and you already show what the implications are of a Jurassic Juramaia in terms of rate shifts. The final conclusion on this can, I think, be improved. In lines 290-293 you say "The Middle Jurassic age of Juramaia suggests unusually rapid rates of evolution at the base of therians and eutherians, followed by a period of exceptionally slow rates of eutherian evolution during the Early Cretaceous". That's all well and good, but I think it can be strengthened by re-emphasising the 50-fold shift in rates (lines 197-199) but also actually telling us about what constitutes a statistically significant deviation from the null hypothesis - ie expectation - of equal rates across the phylogeny. This would be a simple matter of taking the internal branch lengths and comparing them with expected branch lengths with a chi-squared test. This would bring the manuscript back to its title and the lede and justifies the use of the word "unexpected".

I also think it's worth addressing in the discussion the fact that Juramaia is a single specimen of a

single species that then leads to Eomaia being a single specimen of a single species, so we don't really know much about what eutherians are doing in that time at all besides the similarity of these two individuals. The implication of a huge slowdown is interesting, but that kind of long branch is potentially subject to radical change on the introduction of new specimens.

MISSING METHODOLOGY

I've looked through the paper and the supplementary data in detail and cannot find the details of the fossilised birth-death model. This needs to be added to the supplementary information, or, if it is already there, better signposted, because I cannot find it at all.

Did, for instance, you use diversified sampling, and if not, why not, given that most phylogenetic datasets are built on the understanding that this is the case? What, actually, were the priors on taxon age for each of the standard analyses? All we know is that the priors for the Yanliao mammaliaforms when letting their position be a bit more anatomically-driven is a Laplace distribution with 90% within the Jurassic. By implication there are bounds on these mammaliaforms' date priors in the other analyses, but how were they defined? A uniform distribution across the uncertainty in date of the first appearance? If not, what, and why? If the choice of prior matters so much for Juramaia, then we need to know what the prior we are asked to compare the result from a Laplace prior with was.

SECTIONS

The paragraph beginning on Line 118 as far as 121 should not really be in the methods section as it strays into results and discussions. I understand why it ended up there from the point of view of explaining the need for further analysis, but all that's needed is something more like "To further investigate the relative effects of morphological data and topological priors on haramiyid phylogeny...". Otherwise you are revealing some of the results early and end up repeating yourself.

CONSISTENCY

I'd use the same headings for subsections across Methods, Results, and Discussion, for clarity and ease of switching back and forward.

SMALL THINGS, TYPOS, GRAMMAR

Line 45 – double space between 'Patagonia' and 'the'.

Line 47 – double space between 'discussion' and 'is'.

Line 48 – Juramaia sinensis, not sinsensis.

Line 49 – The word should be composing (or another synonym), not comprising. The parts compose the whole; the whole comprises the parts.

Line 50 – double space between 'morphology,' and 'Juramaia'.

Line 57 and elsewhere – 'Tip dating' when referring to the method as a noun. 'Tip-dating' is correct as a compound adjective, e.g. 'a recent tip-dating study' on line 60.

Line 57 – double space between 'investigate' and 'these'

Lines 79-80 – Square brackets seem like an odd choice here. Why not take the reference out of the brackets? On lines 262-263 you have square brackets for the reference within curved brackets.

This could be avoided anyway by saying 'Markov model for variable characters (hereafter Mk_v model)' which also avoids the issue of an unknown abbreviation.

Line 138 – This is the first time BPP has appeared. Some will not recognise the acronym, so expand it. This may also be the case for Line 109 – MYA, although that might be a journal style anyway. See also Line 85 – 'RWTY'; maybe worth noting that it's an R package as I at least assumed it was a method I was unfamiliar with.

Line 141 – 'well resolved', not 'well-resolved'

Review form: Reviewer 2

Recommendation

Major revision is needed (please make suggestions in comments)

Scientific importance: Is the manuscript an original and important contribution to its field?

Good

General interest: Is the paper of sufficient general interest?

Excellent

Quality of the paper: Is the overall quality of the paper suitable?

Good

Is the length of the paper justified?

No

Should the paper be seen by a specialist statistical reviewer?

No

Do you have any concerns about statistical analyses in this paper? If so, please specify them explicitly in your report.

No

It is a condition of publication that authors make their supporting data, code and materials available - either as supplementary material or hosted in an external repository. Please rate, if applicable, the supporting data on the following criteria.

Is it accessible?

N/A

Is it clear?

N/A

Is it adequate?

N/A

Do you have any ethical concerns with this paper?

No

Comments to the Author

See attached file. (See Appendix A)

Decision letter (RSPB-2020-0055.R0)

11-Feb-2020

Dear Mr King:

I am writing to inform you that your manuscript RSPB-2020-0055 entitled "Tip-dating supports novel resolutions of controversial relationships among early mammals" has, in its current form, been rejected for publication in Proceedings B.

This action has been taken on the advice of referees, who have recommended that substantial revisions are necessary. With this in mind we would be happy to consider a resubmission, provided the comments of the referees are fully addressed. However please note that this is not a provisional acceptance. Reanalysis of data will be necessary to satisfy the reviewers and Associate Editor.

Please note that this decision may (or may not) have taken into account confidential comments.

In your revision process, please take a second look at how open your science is; our policy is that *ALL* (maximally inclusive) data involved with the study should be made openly accessible, fully enabling re-use, replication and transparency-- see: <https://royalsociety.org/journals/ethics-policies/data-sharing-mining/>
Insufficient sharing of data can delay or even cause rejection of a paper.

Sincerely,
Professor John Hutchinson, Editor
mailto: proceedingsb@royalsociety.org

Associate Editor
Board Member: 1
Comments to Author:
Dear Authors,

Your manuscript has been seen by two expert reviewers and myself.

Both reviewers find your work of considerable interest, but offer a number of important suggestions that will need to be addressed before your paper will be publishable at Proceedings B.

In particular, Reviewer 2 has numerous recommendations for improvements, chief amongst them the important suggestion to verify the consistency of your results by re-running the analyses on an alternative dataset for Mesozoic mammals.

These suggestions for improvements will demand a more thorough revision of the paper than a basic 'revise' decision can accommodate; as such, I have elected to reject the paper but strongly encourage you to resubmit upon completing the requested revisions.

Ideally, the same referees will receive the paper as on the original submission, so they will be able to judge the soundness of the revised version.

You will have 6 months to submit your revised manuscript, and I would ask that you please submit with the revision a 'tracked changes' document including all edits made since the previous version and a point-by-point letter documenting responses to the reviewers' comments and what changes have been made.

Thank you for considering Proceedings B as a venue for your interesting work; I look forward to seeing a revised version of your paper.

Reviewer(s)' Comments to Author:

Referee: 1

Comments to the Author(s)

King and Beck use up-to-date tip-dating analyses to investigate the relationships among Mesozoic mammals, particularly early eutherians and haramiyids. They conclude that tip-dating alters evolutionary interpretations from other phylogenetic methods by explicitly including stratigraphic data, and note that Juramaia is more similar to later forms than would be expected given its age.

The paper is to my mind strong; my comments below are mostly to do with the presentation of the Juramaia issue and a missing piece of the methodological puzzle.

'UNEXPECTED' MORPHOLOGY OF JURAMAIA

The word "unexpected" comes across as potentially problematic from a philosophical perspective. The basic argument in favour of it would be "Juramaia looks like Cretaceous mammals, but is actually Jurassic. That's unexpected". But what you can and do quantify are branch lengths and evolutionary rate, and you can test for expectation only really on those parameters. In both of the citations given for Juramaia being "unexpectedly advanced", that phrase is only used exactly for Durlstotherium etc. rather than the Juramaia -Eutheria sister relationship by Bi et al., although the sentiments are certainly similar.

However, both the Bi et al. and Meng papers discuss two possible explanations for the similarity between Juramaia and Eomaia. These are (a) an incorrect age estimate for Juramaia or (b) early appearance of eutherian-type dentition and then exceptionally slow rates of evolution. This is the justification for qualitatively noting that something is "unexpected". They also both caution that the fact that Juramaia is represented by a single specimen makes identification of a pattern practically impossible.

I think it's worth noting that the age of the Yanliao Biota is now pretty solidly fixed in the Middle-Late Jurassic - a single adjective when bringing up the biota for the first time would suffice - and you already show what the implications are of a Jurassic Juramaia in terms of rate shifts. The final conclusion on this can, I think, be improved. In lines 290-293 you say "The Middle Jurassic age of Juramaia suggests unusually rapid rates of evolution at the base of therians and eutherians, followed by a period of exceptionally slow rates of eutherian evolution during the Early Cretaceous". That's all well and good, but I think it can be strengthened by re-emphasising the 50-fold shift in rates (lines 197-199) but also actually telling us about what constitutes a statistically significant deviation from the null hypothesis - ie expectation - of equal rates across the phylogeny. This would be a simple matter of taking the internal branch lengths and comparing them with expected branch lengths with a chi-squared test. This would bring the manuscript back to its title and the lede and justifies the use of the word "unexpected".

I also think it's worth addressing in the discussion the fact that Juramaia is a single specimen of a single species that then leads to Eomaia being a single specimen of a single species, so we don't really know much about what eutherians are doing in that time at all besides the similarity of these two individuals. The implication of a huge slowdown is interesting, but that kind of long branch is potentially subject to radical change on the introduction of new specimens.

MISSING METHODOLOGY

I've looked through the paper and the supplementary data in detail and cannot find the details of the fossilised birth-death model. This needs to be added to the supplementary information, or, if it is already there, better signposted, because I cannot find it at all.

Did, for instance, you use diversified sampling, and if not, why not, given that most phylogenetic datasets are built on the understanding that this is the case? What, actually, were the priors on taxon age for each of the standard analyses? All we know is that the priors for the Yanliao mammaliaforms when letting their position be a bit more anatomically-driven is a Laplace distribution with 90% within the Jurassic. By implication there are bounds on these mammaliaforms' date priors in the other analyses, but how were they defined? A uniform distribution across the uncertainty in date of the first appearance? If not, what, and why? If the choice of prior matters so much for Juramaia, then we need to know what the prior we are asked to compare the result from a Laplace prior with was.

SECTIONS

The paragraph beginning on Line 118 as far as 121 should not really be in the methods section as it strays into results and discussions. I understand why it ended up there from the point of view of explaining the need for further analysis, but all that's needed is something more like "To further investigate the relative effects of morphological data and topological priors on haramiyid phylogeny...". Otherwise you are revealing some of the results early and end up repeating yourself.

CONSISTENCY

I'd use the same headings for subsections across Methods, Results, and Discussion, for clarity and ease of switching back and forward.

SMALL THINGS, TYPOS, GRAMMAR

Line 45 – double space between 'Patagonia' and 'the'.

Line 47 – double space between 'discussion' and 'is'.

Line 48 – Juramaia sinensis, not sinsensis.

Line 49 – The word should be composing (or another synonym), not comprising. The parts compose the whole; the whole comprises the parts.

Line 50 – double space between 'morphology,' and 'Juramaia'.

Line 57 and elsewhere – 'Tip dating' when referring to the method as a noun. 'Tip-dating' is correct as a compound adjective, e.g. 'a recent tip-dating study' on line 60.

Line 57 – double space between 'investigate' and 'these'

Lines 79-80 – Square brackets seem like an odd choice here. Why not take the reference out of the brackets? On lines 262-263 you have square brackets for the reference within curved brackets.

This could be avoided anyway by saying 'Markov model for variable characters (hereafter MkV model)' which also avoids the issue of an unknown abbreviation.

Line 138 – This is the first time BPP has appeared. Some will not recognise the acronym, so expand it. This may also be the case for Line 109 – MYA, although that might be a journal style anyway. See also Line 85 – 'RWTY'; maybe worth noting that it's an R package as I at least assumed it was a method I was unfamiliar with.

Line 141 – 'well resolved', not 'well-resolved'

Referee: 2

Comments to the Author(s)

See attached file.

Author's Response to Decision Letter for (RSPB-2020-0055.R0)

See Appendix B.

RSPB-2020-0943.R0

Review form: Reviewer 1 (David M. Grossnickle)

Recommendation

Accept with minor revision (please list in comments)

Scientific importance: Is the manuscript an original and important contribution to its field?

Excellent

General interest: Is the paper of sufficient general interest?

Good

Quality of the paper: Is the overall quality of the paper suitable?

Excellent

Is the length of the paper justified?

Yes

Should the paper be seen by a specialist statistical reviewer?

No

Do you have any concerns about statistical analyses in this paper? If so, please specify them explicitly in your report.

No

It is a condition of publication that authors make their supporting data, code and materials available - either as supplementary material or hosted in an external repository. Please rate, if applicable, the supporting data on the following criteria.

Is it accessible?

Yes

Is it clear?

Yes

Is it adequate?

Yes

Do you have any ethical concerns with this paper?

No

Comments to the Author

See attached file.

Decision letter (RSPB-2020-0943.R0)

11-May-2020

Dear Mr King

I am pleased to inform you that your manuscript RSPB-2020-0943 entitled "Tip dating supports novel resolutions of controversial relationships among early mammals" has been accepted for publication in Proceedings B. Congratulations!!

The referee(s) have recommended publication, but also suggest some minor revisions to your manuscript. Therefore, I invite you to respond to the referee(s)' comments and revise your manuscript. Because the schedule for publication is very tight, it is a condition of publication that you submit the revised version of your manuscript within 7 days. If you do not think you will be able to meet this date please let us know.

Sincerely,

Dr John Hutchinson, Editor

Associate Editor

Board Member

Comments to Author:

Dear Authors,

Thank you for your revised submission to Proceedings B. Your paper has been seen by the first round reviewer who had the most extensive first round comments, and as you'll see they are very satisfied with the modifications you have made to your study, and recognise your study's significance.

Importantly, they do make a number of suggested revisions, which I encourage you to fully consider. These suggested revisions are minor and I doubt they will take too much effort to address.

Thanks once again for submitting your interesting work to Proceedings B.

Author's Response to Decision Letter for (RSPB-2020-0943.R0)

See Appendix C.

Decision letter (RSPB-2020-0943.R1)

13-May-2020

Dear Mr King

I am pleased to inform you that your manuscript entitled "Tip dating supports novel resolutions of controversial relationships among early mammals" has been accepted for publication in Proceedings B.

Your article has been estimated as being 7 pages long. Our Production Office will be able to confirm the exact length at proof stage.

Open Access

Paper charges

Sincerely,
Editor, Proceedings B
<mailto:proceedingsb@royalsociety.org>

Appendix A

General comments

There's ongoing, contentious debate on allotherian relationships. Because allotherians likely evolved near the crown mammalian node, resolving this issue is especially important for understanding the origins and early evolution of mammals. Thus, the study by King and Beck is very relevant and should be of interest to a broad readership. I have no major concerns with the tip-dating methods or results. And the primary result (i.e. haramiyids are diphyletic) makes intuitive sense and offers somewhat of a compromise for the competing hypotheses. The second major conclusion of the paper is that Juramaia is morphologically very similar to much younger taxa, suggesting that therians evolved very rapidly early in their history and then experienced a long period of stasis (or Juramaia's age is incorrect). The analyses offer interesting insights on this topic.

I don't have any critical concerns with the paper. However, I believe it would benefit greatly from two major additions, which I outline here.

To make the paper easier to follow for non-specialists, I recommend adding an introductory figure with simplified phylogenies that summarizes the competing allotherian hypotheses. I envision something like Figure 1 from Cifelli & Davis 2013 (*Nature, News & Views*) or Figure 4A & B from Luo et al. 2015 (PNAS). Your figure could also include a third hypothesis in which haramiyids are diphyletic. Allotherian lineages could be colored green to stay consistent with Figure 1. You could also consider adding a panel to the figure that highlights the considerable age difference between Juramaia and Jehol Biota therians.

My second major suggestion is to perform supplemental analyses using a character matrix from a recent Jin Meng paper. The two sides of the allotherian debate are driven by Jin Meng and Zhe-Xi Luo, and you have analyses that use a matrix from Luo (Huttenlocker et al. 2018), but you don't have equivalent analyses based on a Meng matrix. (Meng's matrices are derived from earlier Luo matrices, but scoring of many characters varies considerably, especially for allotherians.) Analyses using the Krause et al. 2014 matrix are fine, especially because it includes more multituberculates than the Meng and Luo matrices. Krause et al. followed Meng's scorings of haramiyid dental characters for their primary analysis, so their matrix indirectly represents Meng's interpretation of allotherians. However, many new allotherians have been published since 2014, and the only euharamiyidan in Krause et al. is *Arboroharamiya*. To truly account for both sides of the allotherian debate, it would be best to include a recent matrix from Meng. I recommend Wang et al. 2019 (*Nature*) because the described taxon is an allotherian (*Jeholbaatar*, a multituberculate). Meng's scoring of euharamiyid dental characters will likely unite multituberculates and euharamiyids in your analyses (as seen in the Krause-matrix results), but if you continue to see evidence of haramiyid diphyly then it will provide even stronger support for your results.

Personally, I agree with Luo's scorings of haramiyid dental characters – I think he's made a more compelling case for haramiyid occlusion (following Jenkins 1997 and Butler 2000) using CT scans and occlusal reconstructions (e.g. see Fig. S8 in Luo et al. 2015 PNAS). So, I think you should continue to focus on results derived from the Huttenlocker matrix. But because of the debate on the topic, as well as tangent disagreements on non-dental characters, it is worth including analyses based on the Wang et al. 2019 matrix (or a similar variant).

Your results offer somewhat of a compromise on the debate over haramiyids, and you could consider re-framing some of the text to focus on this. Recovering euharamiyids as crown

mammals supports Meng's view, and recovering Haramiyavia+Thomasia as stem mammals is consistent with Luo's view. You could consider putting greater emphasis on the current debate (e.g. by adding more text and citations to the first paragraph of the Introduction) and providing more discussion on how the results may serve as a compromise between the two sides.

Comments on Juramaia results

I'm concerned that the Juramaia results are influenced by sampling issues. The other Yanliao taxa are members of clades that are well-represented in the Jurassic, whereas Juramaia is the only therian in the Jurassic. If more Jurassic therians were discovered, would it drastically change the results of this study by introducing new morphological variation and altering branch lengths? One potential way to test this idea is to simulate Juramaia-like scenarios in other clades. For instance, you could remove Kuehneodon and set the age of the plagiaulacid multituberculates to ca. 126 Ma (to match the Jehol), and then predict the age of Rugosodon (the earliest multituberculate in the study) based on morphology. This could serve as somewhat of a sensitivity analysis to examine the influence of sampling on the Juramaia results.

Some authors (e.g. Sweetman et al. 2017 Acta Palaeo Polon) have posited that Juramaia is a stem therian rather than a eutherian. If this is the case, I doubt it'd alter your conclusions because there'd still be a very slow rate of morphological between Juramaia and other early therians. But it'd significantly alter some branch lengths, and thus likely change some results. For example, this topological change would likely lengthen the branch leading to eutherians, and thus in Figure 3a I'd expect Branch 2 to have much slower rates of evolution. I think this topic should be addressed at some point (even if it's in the supplement) because many readers may wonder what influence the phylogenetic position of Juramaia has on results.

Sinodelphys from the Jehol Biota (previously thought to be the oldest metatherian) is now believed to be a eutherian (Bi et al. 2018 Nature). Incorporating this change into analyses is probably beyond the scope of this study, but it's another variable that could influence the Juramaia/therian results. The authors could note whether they think that moving Sinodelphys to Eutheria would alter branch lengths and evolutionary rates.

Although the Huttenlocker (i.e. Luo) and Krause matrices include therians, the characters are primarily designed to investigate relationships in earlier groups such as docodonts, allotherians, and eutriconodonts. I wonder if this results in artificially slow evolutionary rates along early therian branches because there are few characters that help to distinguish these lineages. (I think this is partly why Close et al. 2015 report very slow rates of evolution in the Late Cretaceous despite the ecological diversification of taxa (Grossnickle & Newham 2016 Proc B, Wilson et al. 2016 Nature Com.)). It might be something for the authors to consider. I'd be interested to see if therian-focused matrices (e.g. Bi et al. 2018 Nature) generated different results for evolutionary rates along the branches of interest.

Specific comments

Lines 9, 12, and 179: I don't think that "problematic" is the best term for describing the taxa. It's problematic that there are different interpretations of the fossils, but I'd be hesitant to state that haramiyids and Juramaia themselves are problematic. Juramaia has unexpected morphology given its age, but I think this makes it especially interesting (or simply raises

questions about its provenance). For Line 12, maybe instead state “A second taxon of special interest is ...”.

Lines 17-18, 143-145, and 259: Although haramiyid diphyly isn't reported in the main text of any study, a supplemental analysis in Luo et al. 2015 (PNAS) recovered diphyly of haramiyids using a re-scored version of the Krause et al. 2014 matrix (see pages 19 and 65 of their supplemental file). Consistent with some of your Krause-matrix results, Luo et al. recovered *Arboroharamiya* (the only euharamiyidan) as sister to multituberculates. You could still emphasize that the Huttenlocker-matrix results are novel (showing three unique allotherian lineages), but I would revise your statements (Lines 17-18, 143-145, and 259) and acknowledge the Luo et al. 2015 analysis.

Lines 18-19: You state that *Haramiyavia* and *Thomasia* form a clade with tritylodontids, but you later devote a paragraph (Lines 270-285) to explaining why this relationship is unlikely. I agree with your skepticism on this relationship and recommend you remove the mention of it here, or rephrase the statement so that it's less assertive.

Line 31: Consider erasing “or not the so-called”.

Lines 31-34: I recommend you note that allotherians are extinct, and possibly provide additional background information on allotherians if there's space. Many readers may be unfamiliar with the group.

Line 36: In addition to refs 3 and 8, ref 36 (Luo et al. 2017, *Nature*) can also be cited here. They performed a phylogenetic analysis in their supplement. I also recommend citing Zhou et al. 2013 (*Nature*, on *Megaconus*). Zheng et al. 2013 (ref 4) and Zhou et al. 2013 were published in the same issue of *Nature*, and their differing conclusions ignited the current debate.

Line 52: I recommend “ca. 126” (circa) instead of “~126”.

Lines 57-60: This section does not clearly introduce tip-dating approaches. Just stating “Tip-dating” alone (Line 57) seems vague. Consider expanding the phrase (e.g. to “Tip-dating phylogenetic methods” or “Tip-dating Bayesian approaches for phylogenetic inference”) and briefly mentioning how tip-dating approaches differ from more traditional phylogenetic methods. Similarly, “method” on Line 59 is vague. Maybe expand it to “phylogenetic method” or re-phrase the sentence.

Line 73: Maybe I missed it, but what's your source(s) for stratigraphic ages of taxa? Is it the Paleobiology Database? Or primary literature? This should be mentioned in the Methods.

Lines 163–178: This is an important section that deserves further discussion. The Huttenlocker et al. (2018) matrix conforms with Luo's scoring of allotherians, whereas the Krause et al. (2014) largely conforms with Meng's scoring of allotherians. The fact that tip-dating does not resolve these two competing interpretations of allotherian anatomy/phylogenetic scoring (i.e.,

there is still support for a monophyletic allotheria when taxa are scored following Meng's interpretation) should be recognized. I predict you will get a similar (if not stronger) result if you use the Wang et al. (2019) matrix. I think it's fine to emphasize your Huttenlocker matrix result, especially because it presents a compelling alternative hypothesis for allotherian relationships, but it's important to also address the fact that differential scoring of key taxa and characters is still having a major effect on the results.

Lines 235–238: This could be a good point in the manuscript to incorporate my previous comment.

Lines 246-249: You should note that this result is based only on the Huttenlocker-matrix results, because the Krause-matrix results suggest monophyly or diphyly of Allotheria. Although there is compelling evidence for diphyly of Haramiyids, you should be quick to acknowledge (here and throughout the manuscript) that there's considerable variation and uncertainty in the tree topologies.

Lines 246-252: The argument that placentals and marsupials have occasionally converged (in a broad sense) on allotherian-like dental traits has been frequently brought up in the literature (e.g., Simpson 1933, J Mamm; Lazzari et al. 2010, J Mamm Evol), so pointing this out is justified. However, I wouldn't call this "unsurprising". The dental convergence with placentals (e.g., mureoid rodent molars), marsupials (e.g., blade-like cheek teeth), and even tritylodontids are only similar at a very superficial level (i.e., there has never been a strong case made for these characteristics being homologous). In contrast, it has been argued that the dentitions of Haramiyidans and multituberculates are homologous, not homoplastic (Butler 2000, Acta Palaeo Polon; Butler & Hooker 2005, Acta Palaeo Polon; Meng papers). I think it's fine to touch on the fact that these dental traits are possibly convergent. However, I recommend also acknowledging the studies that argue a contrary point.

And although it's briefly mentioned in the Introduction, I'd give an example or two of how the dentitions are similar among allotherian groups (and rodents and polydolopimorphians). More broadly, I think that the paper would benefit from more discussion on the morphological traits of the early mammal groups being discussed in the paper. I know that Proc B has a short word limit, but if there's space I recommend an expanded discussion on topics such as dental characters and ear-jaw evolution, especially in allotherians.

Lines 270-285: I agree with this discussion and highly doubt that Haramiyavia+Thomasia are sister to tritylodontids. As you note, there are many traits that differentiate Haramiyavia+Thomasia from tritylodontids. But this highlights an underlying concern – the Huttenlocker/Luo and Krause matrix characters were generated to investigate relationships of mammaliaforms, not cynodonts. (This is analogous to my above comments on therians.) There are relatively few cynodont-specific characters in those matrices. And the scarcity of sampled cynodont taxa in the matrices may help to generate the long branches and 'attract' Haramiyavia+Thomasia. A cynodont-focused matrix might be more likely to separate tritylodontids from Haramiyavia+Thomasia. I recommend acknowledging this concern in the text.

Line 275: Although the chewing stroke was likely predominantly orthal in Haramiyavia, Butler (2000) suggested a significant palinal component in the chewing stroke of Thomasia.

Lines 288–293: This result might also suggest that Juramaia might not be Jurassic in age, and instead is Early Cretaceous, but you seem hesitant to bring it up (maybe previous commenters/reviewers on this manuscript have dismissed that possibility?). Personally, I think it's worth mentioning as a possibility, even if you doubt it's the case. I've also seen Juramaia's age questioned in Meng 2014 (Nat Sci Rev), and there might be other references that raise similar concerns.

Figure 1: I recommend labeling key nodes, such as Mammaliaformes, Mammalia, Theria, and Eutheria. This will be especially important for readers to better understand which branches are being highlighted in Figure 3 (see comment below).

Figure 2: I recommend flipping the x-axis so that younger ages are to the right, especially because to stay consistent with Figure 1.

Figure 3: The results in this figure may be difficult to interpret because the branches of interest are not displayed. Consider labeling the three branches (and relevant nodes) in Figure 1, and citing Figure 1 in the Figure 3 caption.

Appendix B

Reviewer 1

Comments to the Author(s)

King and Beck use up-to-date tip-dating analyses to investigate the relationships among Mesozoic mammals, particularly early eutherians and haramiyids. They conclude that tip dating alters evolutionary interpretations from other phylogenetic methods by explicitly including stratigraphic data, and note that *Juramaia* is more similar to later forms than would be expected given its age.

The paper is to my mind strong; my comments below are mostly to do with the presentation of the *Juramaia* issue and a missing piece of the methodological puzzle.

'UNEXPECTED' MORPHOLOGY OF JURAMAIA

The word “unexpected” comes across as potentially problematic from a philosophical perspective. The basic argument in favour of it would be “*Juramaia* looks like Cretaceous mammals, but is actually Jurassic. That’s unexpected”. But what you can and do quantify are branch lengths and evolutionary rate, and you can test for expectation only really on those parameters. In both of the citations given for *Juramaia* being “unexpectedly advanced”, that phrase is only used exactly for *Durlstotherium* etc. rather than the *Juramaia* –*Eutheria* sister relationship by Bi et al., although the sentiments are certainly similar. However, both the Bi et al. and Meng papers discuss two possible explanations for the similarity between *Juramaia* and *Eomaia*. These are (a) an incorrect age estimate for *Juramaia* or (b) early appearance of eutherian-type dentition and then exceptionally slow rates of evolution. This is the justification for qualitatively noting that something is “unexpected”. They also both caution that the fact that *Juramaia* is represented by a single specimen makes identification of a pattern practically impossible.

Authors’ response: We realise that the reported age of *Juramaia* has been questioned in several papers, but none of these paper shave presented evidence to support this conclusion besides qualitative remarks on its close morphological resemblance to the Jehol eutherians. We have quantified this resemblance for the first time by showing that, when the age of *Juramaia* is allowed to vary, our morphological clock analysis estimates the age of *Juramaia* to be similar to the Jehol eutherians. However, we emphasise that this does not show that the age of *Juramaia* is incorrect – Luo et al. (2011) clearly state that *Juramaia* is from the Daxishan/Daxigou site of Jianchang County of Liaoning Province, China, and the age of this locality is tightly constrained as ~160 MYA. In our opinion, the only way to show that the age of *Juramaia* is incorrect is by showing that the fossil does not come from that locality, and this is not something that our paper addresses. We therefore must assume that the reported age of *Juramaia* is correct. Our analysis simply shows that the known morphology of *Juramaia* is not what we would expect given its reported age and the assumptions of our analysis (model priors etc.). For this reason, we consider “unexpected” to be the most appropriate, neutral term for the situation regarding *Juramaia*. However, we have now taken care to put the word ‘unexpected’ in quotation marks throughout the paper.

I think it’s worth noting that the age of the Yanliao Biota is now pretty solidly fixed in the Middle-Late Jurassic – a single adjective when bringing up the biota for the first time would suffice – and you already show what the implications are of a Jurassic *Juramaia* in terms of rate shifts. The final conclusion on this can, I think, be improved. In lines 290-293 you say “The Middle Jurassic age of *Juramaia* suggests unusually rapid rates of evolution at the base of therians and eutherians, followed

by a period of exceptionally slow rates of eutherian evolution during the Early Cretaceous". That's all well and good, but I think it can be strengthened by re-emphasising the 50-fold shift in rates (lines 197-199) but also actually telling us about what constitutes a statistically significant deviation from the null hypothesis – ie expectation – of equal rates across the phylogeny. This would be a simple matter of taking the internal branch lengths and comparing them with expected branch lengths with a chi-squared test. This would bring the manuscript back to its title and the lede and justifies the use of the word "unexpected".

Authors' response: We have reiterated the 50-fold decrease: "...followed by a 50-fold rate decrease and a period of exceptionally slow eutherian morphological evolution during the Early Cretaceous" Our test of the age of *Juramaia* using relaxed priors (figure 3, formerly figure 2) is the most intuitive and accurate way to test this 'expectation' and we also show the effect on branch rates in figure S13 (formerly figure 3). The reviewer states that the null hypothesis is equal rates across the phylogeny, but this is incorrect. We have used a relaxed clock model (the uncorrelated lognormal clock model) that specifically allows rates to vary between branches according to an underlying lognormal distribution. Indeed, this is exactly what we see – even when the age of *Juramaia* is allowed to vary, we still see considerable rate heterogeneity, between <0.01 and 0.08 transitions/character/million years (figure S10). As a result, a chi-squared test with a null expectation of equal rates is not appropriate here.

I also think it's worth addressing in the discussion the fact that *Juramaia* is a single specimen of a single species that then leads to *Eomaia* being a single specimen of a single species, so we don't really know much about what eutherians are doing in that time at all besides the similarity of these two individuals. The implication of a huge slowdown is interesting, but that kind of long branch is potentially subject to radical change on the introduction of new specimens.

Authors' response: We fully agree that there is a limit to how much we can conclude based on single specimens. However, even single specimens provide robust minimum age constraints on the evolution of particular features in specific lineages (assuming they can be correctly placed in the phylogeny). So, even though it's a single specimen, *Juramaia* shows that the morphology typical of known Early Cretaceous eutherians had originated by the Middle-Late Jurassic. Even if we find lots of eutherian fossils that fall on the long branch leading from *Juramaia* to other eutherians, it would not alter the fact that *Eomaia*, *Ambolestes* and other currently known Early Cretaceous eutherians have experienced a particularly low rate of morphological evolution.

To address the reviewer's point, we have added the following sentences to the end of the discussion:

"However, this result requires further scrutiny, as it largely driven by two taxa, both of which are known from single specimens: *Juramaia* and *Eomaia*. The highly incomplete record of early eutherians makes it difficult to reach robust conclusions regarding the macroevolution of the group, and these may change with future discoveries."

MISSING METHODOLOGY

I've looked through the paper and the supplementary data in detail and cannot find the details of the fossilised birth-death model. This needs to be added to the supplementary information, or, if it is already there, better signposted, because I cannot find it at all. Did, for instance, you use diversified sampling, and if not, why not, given that most phylogenetic datasets are built on the understanding

that this is the case? What, actually, were the priors on taxon age for each of the standard analyses? All we know is that the priors for the Yanliao mammaliaforms when letting their position be a bit more anatomically-driven is a Laplace distribution with 90% within the Jurassic. By implication there are bounds on these mammaliaforms' date priors in the other analyses, but how were they defined? A uniform distribution across the uncertainty in date of the first appearance? If not, what, and why? If the choice of prior matters so much for Juramaia, then we need to know what the prior we are asked to compare the result from a Laplace prior with was.

Authors' response: Diversified sampling is only relevant when there is a sample of extant taxa. We now state explicitly how taxon age priors were specified by adding the following sentence to our methods:

"Tip dates were assigned uniform priors across the range of uncertainty for each taxon".

The supplementary information now includes the age ranges for every taxon including references. We also include a section in the supplementary information with the full details of the analysis.

SECTIONS

The paragraph beginning on Line 118 as far as 121 should not really be in the methods section as it strays into results and discussions. I understand why it ended up there from the point of view of explaining the need for further analysis, but all that's needed is something more like "To further investigate the relative effects of morphological data and topological priors on haramiyid phylogeny...". Otherwise you are revealing some of the results early and end up repeating yourself.

Authors' response: We have changed this to:

"To test the effect of taxon age on the phylogenetic position of haramiyidans, we ran an analysis..."

CONSISTENCY

I'd use the same headings for subsections across Methods, Results, and Discussion, for clarity and ease of switching back and forward.

Authors' response: changed as requested

SMALL THINGS, TYPOS, GRAMMAR

Line 45 – double space between 'Patagonia' and 'the'.

Authors' response: changed as requested

Line 47 – double space between 'discussion' and 'is'.

Authors' response: changed as requested

Line 48 – Juramaia sinensis, not sinsensis.

Authors' response: changed as requested

Line 49 – The word should be composing (or another synonym), not comprising. The parts compose the whole; the whole comprises the parts.

Authors' response: changed as requested

Line 50 – double space between 'morphology,' and 'Juramaia'.

Authors' response: changed as requested

Line 57 and elsewhere – 'Tip dating' when referring to the method as a noun. 'Tip-dating' is correct as a compound adjective, e.g. 'a recent tip-dating study' on line 60.

Authors' response: changed throughout as requested

Line 57 – double space between 'investigate' and 'these'

Authors' response: changed as requested

Lines 79-80 – Square brackets seem like an odd choice here. Why not take the reference out of the brackets? On lines 262-263 you have square brackets for the reference within curved brackets. This could be avoided anyway by saying 'Markov model for variable characters (hereafter Mkv model)' which also avoids the issue of an unknown abbreviation.

Authors' response: Changed to:

"The Markov model for variable characters (hereafter Mkv) was used"

Line 138 – This is the first time BPP has appeared. Some will not recognise the acronym, so expand it.

Authors' response: changed as requested

This may also be the case for Line 109 – MYA, although that might be a journal style anyway.

Authors' response: We have changed MYA to Ma throughout to follow journal style

See also Line 85 – 'RWTY'; maybe worth noting that it's an R package as I at least assumed it was a method I was unfamiliar with.

Authors' response: Changed to

"was confirmed using the R package RWTY"

Line 141 – 'well resolved', not 'well-resolved'

Authors' response: changed as requested

Reviewer 2

General comments

There's ongoing, contentious debate on allotherian relationships. Because allotherians likely evolved near the crown mammalian node, resolving this issue is especially important for understanding the origins and early evolution of mammals. Thus, the study by King and Beck is very relevant and should be of interest to a broad readership. I have no major concerns with the tip-dating methods or results. And the primary result (i.e. haramiyids are diphyletic) makes intuitive sense and offers somewhat of a compromise for the competing hypotheses. The second major conclusion of the paper is that Juramaia is morphologically very similar to much younger taxa, suggesting that therians evolved very rapidly early in their history and then experienced a long period of stasis (or Juramaia's age is incorrect). The analyses offer interesting insights on this topic.

I don't have any critical concerns with the paper. However, I believe it would benefit greatly from two major additions, which I outline here.

To make the paper easier to follow for non-specialists, I recommend adding an introductory figure with simplified phylogenies that summarizes the competing allotherian hypotheses. I envision something like Figure 1 from Cifelli & Davis 2013 (Nature, News & Views) or Figure 4A & B from Luo et al. 2015 (PNAS). Your figure could also include a third hypothesis in which haramiyids are diphyletic. Allotherian lineages could be colored green to stay consistent with Figure 1.

Authors' response: We have now added this new figure 1

You could also consider adding a panel to the figure that highlights the considerable age difference between Juramaia and Jehol Biota therians.

Authors' response: Figure 2 (previously figure 1) highlights this difference, so we have referenced that. We feel that adding another panel to figure 1 would be redundant, and would take up unnecessary space.

My second major suggestion is to perform supplemental analyses using a character matrix from a recent Jin Meng paper. The two sides of the allotherian debate are driven by Jin Meng and Zhe-Xi Luo, and you have analyses that use a matrix from Luo (Huttenlocker et al. 2018), but you don't have equivalent analyses based on a Meng matrix. (Meng's matrices are derived from earlier Luo matrices, but scoring of many characters varies considerably, especially for allotherians.) Analyses using the Krause et al. 2014 matrix are fine, especially because it includes more multituberculates than the Meng and Luo matrices. Krause et al. followed Meng's scorings of haramiyid dental characters for their primary analysis, so their matrix indirectly represents Meng's interpretation of allotherians. However, many new allotherians have been published since 2014, and the only euharamiyidan in Krause et al. is Arboroharamiya. To truly account for both sides of the allotherian debate, it would be best to include a recent matrix from Meng. I recommend Wang et al. 2019 (Nature) because the described taxon is an allotherian (Jeholbaatar, a multituberculate). Meng's scoring of euharamiyid dental characters will likely unite multituberculates and euharamiyids in your analyses (as seen in the Krause-matrix results), but if you continue to see evidence of haramiyid diphyly then it will provide even stronger support for your results. Personally, I agree with Luo's scorings of haramiyid dental characters – I think he's made a more compelling case for haramiyid occlusion (following Jenkins 1997 and Butler 2000) using CT scans and occlusal reconstructions (e.g. see Fig. S8 in Luo et al. 2015 PNAS). So, I think you should continue to focus on results derived from the Huttenlocker matrix. But because of the debate on the topic, as well as tangent disagreements on non-dental characters, it is worth including analyses based on the Wang et al. 2019 matrix (or a similar variant).

Authors' response: We have analysed the Wang et al. dataset and now include it in the results and as figure S6. As predicted , this analysis recovers haramiyidan diphyly.

Your results offer somewhat of a compromise on the debate over haramiyids, and you could consider re-framing some of the text to focus on this. Recovering euharamiyids as crown mammals supports Meng's view, and recovering Haramiyavia+Thomasia as stem mammals is consistent with Luo's view. You could consider putting greater emphasis on the current debate (e.g. by adding more text and citations to the first paragraph of the Introduction) and providing more discussion on how the results may serve as a compromise between the two sides.

Authors' response: We thank the reviewer for this perceptive point. We have added the following text to our discussion

“In some ways, our results represent a compromise between differing views on whether haramiyidans are crown- or stem-mammals: euharamiyidans fall within the crown-clade, whereas *Haramiyavia+Thomasia* fall outside.”

Comments on Juramaia results

I'm concerned that the Juramaia results are influenced by sampling issues. The other Yanliao taxa are members of clades that are well-represented in the Jurassic, whereas Juramaia is

the only therian in the Jurassic. If more Jurassic therians were discovered, would it drastically change the results of this study by introducing new morphological variation and altering branch lengths? One potential way to test this idea is to simulate Juramaia-like scenarios in other clades. For instance, you could remove *Kuehneodon* and set the age of the plagiaulacid multituberculates to ca. 126 Ma (to match the Jehol), and then predict the age of *Rugosodon* (the earliest multituberculate in the study) based on morphology. This could serve as somewhat of a sensitivity analysis to examine the influence of sampling on the Juramaia results.

Authors' response: We have tested this by removing *Kuehneodon* and plagiaulacids, and predicting the age of *Rugosodon*. This is included as figure S11. This analysis does suggest that the *Juramaia* result could be partially (but not entirely) driven by sampling, and we now discuss this.

Some authors (e.g. Sweetman et al. 2017 Acta Palaeo Polon) have posited that Juramaia is a stem therian rather than a eutherian. If this is the case, I doubt it'd alter your conclusions because there'd still be a very slow rate of morphological between Juramaia and other early therians. But it'd significantly alter some branch lengths, and thus likely change some results. For example, this topological change would likely lengthen the branch leading to eutherians, and thus in Figure 3a I'd expect Branch 2 to have much slower rates of evolution. I think this topic should be addressed at some point (even if it's in the supplement) because many readers may wonder what influence the phylogenetic position of Juramaia has on results. *Sinodelphys* from the Jehol Biota (previously thought to be the oldest metatherian) is now believed to be a eutherian (Bi et al. 2018 Nature). Incorporating this change into analyses is probably beyond the scope of this study, but it's another variable that could influence the Juramaia/therian results. The authors could note whether they think that moving *Sinodelphys* to Eutheria would alter branch lengths and evolutionary rates. Although the Huttenlocker (i.e. Luo) and Krause matrices include therians, the characters are primarily designed to investigate relationships in earlier groups such as docodonts, allotherians, and eutriconodonts. I wonder if this results in artificially slow evolutionary rates along early therian branches because there are few characters that help to distinguish these lineages. (I think this is partly why Close et al. 2015 report very slow rates of evolution in the Late Cretaceous despite the ecological diversification of taxa (Grossnickle & Newham 2016 Proc B, Wilson et al. 2016 Nature Com.)). It might be something for the authors to consider. I'd be interested to see if therian-focused matrices (e.g. Bi et al. 2018 Nature) generated different results for evolutionary rates along the branches of interest.

Authors' response: We have added the following passage at the end of our discussion:

"This result requires further scrutiny due to low sampling and phylogenetic uncertainty of early therian mammals. Our result is largely driven by two taxa, both of which are known from single specimens: Juramaia and Eomaia. The highly incomplete record of early eutherians [16] makes it difficult to reach robust conclusions regarding the macroevolution of the group, and these may change with future discoveries. Juramaia has also been considered to be a stem therian by some authors [44], a phylogenetic position that would be more consistent with its age. Finally, Sinodelphys has recently been proposed to be a eutherian rather than a metatherian [16]. If this is the case, it could alter branch length estimates, and influence recovered patterns of early eutherian evolution."

Specific comments

Lines 9, 12, and 179: I don't think that "problematic" is the best term for describing the taxa.

It's problematic that there are different interpretations of the fossils, but I'd be hesitant to state that haramiyids and Juramaia themselves are problematic. Juramaia has unexpected morphology given its age, but I think this makes it especially interesting (or simply raises questions about its provenance). For Line 12, maybe instead state "A second taxon of special interest is ...".

Authors' response: The word problematic has been removed in each case

Lines 17-18, 143-145, and 259: Although haramiyid diphyly isn't reported in the main text of any study, a supplemental analysis in Luo et al. 2015 (PNAS) recovered diphyly of haramiyids using a re-scored version of the Krause et al. 2014 matrix (see pages 19 and 65 of their supplemental file) Consistent with some of your Krause-matrix results, Luo et al. recovered Arboroharamiya (the only euharamiyidan) as sister to multituberculates. You could still emphasize that the Huttenlocker-matrix results are novel (showing three unique allotherian lineages), but I would revise your statements (Lines 17-18, 143-145, and 259) and acknowledge the Luo et al. 2015 analysis.

Authors' response: We have modified the abstract as follows :

"Tip dating firmly rejects a monophyletic Allotheria (multituberculates and haramiyidans), which are split into three independent clades, a result not found in any previous analysis"

In the results and discussion, we deleted the sentences referred to by the reviewer, as they are redundant

Lines 18-19: You state that Haramiyavia and Thomasia form a clade with tritylodontids, but you later devote a paragraph (Lines 270-285) to explaining why this relationship is unlikely. I agree with your skepticism on this relationship and recommend you remove the mention of it here, or rephrase the statement so that it's less assertive.

Authors' response: We have deleted this from the abstract

Line 31: Consider erasing "or not the so-called".

Authors' response: changed as requested

Lines 31-34: I recommend you note that allotherians are extinct, and possibly provide additional background information on allotherians if there's space. Many readers may be unfamiliar with the group.

Authors' response: We have modified the text so that it now reads as follows:

"Allotherians are an extinct group of mammaliaforms, primarily known from the Mesozoic, that are currently the subject of conflicting phylogenetic hypotheses (figure 1)."

Line 36: In addition to refs 3 and 8, ref 36 (Luo et al. 2017, Nature) can also be cited here. They performed a phylogenetic analysis in their supplement. I also recommend citing Zhou et al. 2013 (Nature, on Megaconus). Zheng et al. 2013 (ref 4) and Zhou et al. 2013 were published in the same issue of Nature, and their differing conclusions ignited the current debate.

Authors' response: we have added citations to both Zhou et al. 2013 and Luo et al. 2017, as suggested

Line 52: I recommend “ca. 126” (circa) instead of “~126”.

Authors’ response: changed as requested

Lines 57-60: This section does not clearly introduce tip-dating approaches. Just stating “Tipdating” alone (Line 57) seems vague. Consider expanding the phrase (e.g. to “Tip-dating phylogenetic methods” or “Tip-dating Bayesian approaches for phylogenetic inference”) and briefly mentioning how tip-dating approaches differ from more traditional phylogenetic methods. Similarly, “method” on Line 59 is vague. Maybe expand it to “phylogenetic method” or re-phrase the sentence.

Authors’ response: we have rephrased line 57 to:

“Tip-dated phylogenetic methods which include morphological and stratigraphic data in a single coherent analytical framework, are a promising avenue to investigate these issues”

We have also added the following to line 59: “and incorporating stratigraphic data into phylogenetic analysis”

Line 73: Maybe I missed it, but what’s your source(s) for stratigraphic ages of taxa? Is it the Paleobiology Database? Or primary literature? This should be mentioned in the Methods.

Authors’ response: We have now included a fully referenced list of taxon age ranges in the supplementary information (this was mistakenly omitted in the first submission), along with an explanation of the approach taken.

Lines 163–178: This is an important section that deserves further discussion. The Huttenlocker et al. (2018) matrix conforms with Luo’s scoring of allotherians, whereas the Krause et al. (2014) largely conforms with Meng’s scoring of allotherians. The fact that tip-dating does not resolve these two competing interpretations of allotherian anatomy/phylogenetic scoring (i.e., there is still support for a monophyletic allotheria when taxa are scored following Meng’s interpretation) should be recognized. I predict you will get a similar (if not stronger) result if you use the Wang et al. (2019) matrix. I think it’s fine to emphasize your Huttenlocker matrix result, especially because it presents a compelling alternative hypothesis for allotherian relationships, but it’s important to also address the fact that differential scoring of key taxa and characters is still having a major effect on the results.

Lines 235–238: This could be a good point in the manuscript to incorporate my previous comment.

Authors’ response: We have included a new paragraph, as follows:

“The recovered phylogenetic relationships of allotherians depend on both the dataset and the method used. Use of tip-dated methods invariably push the results towards splitting up the allotherians, but the extent of this depends on the data matrix. For the datasets from Krause et al. and Wang et al., which originally recovered allotherian monophyly (Fig 1, topology 1), tip dating leads to increased support for two independent lineages (Fig. 1, topology 2b). For the dataset from Huttenlocker et al., which originally recovered separate haramiyidans and multituberculates (topology 2a), tip dating decisively supports three independent lineages (topology 3).”

Lines 246-249: You should note that this result is based only on the Huttenlocker-matrix results, because the Krause-matrix results suggest monophyly or diphyly of Allotheria.

Although there is compelling evidence for diphyly of haramiyids, you should be quick to acknowledge (here and throughout the manuscript) that there's considerable variation and uncertainty in the tree topologies.

Authors' response: we have changed this to:

"The results of our tip-dating analysis of the Huttenlocker et al. dataset suggest..."

Lines 246-252: The argument that placentals and marsupials have occasionally converged (in a broad sense) on allotherian-like dental traits has been frequently brought up in the literature (e.g., Simpson 1933, J Mamm; Lazzari et al. 2010, J Mamm Evol), so pointing this out is justified. However, I wouldn't call this "unsurprising". The dental convergence with placentals (e.g., mureoid rodent molars), marsupials (e.g., blade-like cheek teeth), and even tritylodontids are only similar at a very superficial level (i.e., there has never been a strong case made for these characteristics being homologous). In contrast, it has been argued that the dentitions of haramiyidans and multituberculates are homologous, not homoplastic (Butler 2000, Acta Palaeo Polon; Butler & Hooker 2005, Acta Palaeo Polon; Meng papers). I think it's fine to touch on the fact that these dental traits are possibly convergent. However, I recommend also acknowledging the studies that argue a contrary point. And although it's briefly mentioned in the Introduction, I'd give an example or two of how the dentitions are similar among allotherian groups (and rodents and polydolopimorphians). More broadly, I think that the paper would benefit from more discussion on the morphological traits of the early mammal groups being discussed in the paper. I know that Proc B has a short word limit, but if there's space I recommend an expanded discussion on topics such as dental characters and ear-jaw evolution, especially in allotherians.

Authors' response: We agree with the reviewer that our statement "The finding that the dental resemblances between allotherians is the result of convergence is perhaps unsurprising, as broadly similar combinations of dental features has also evolved in placental mammals (e.g. rodents) and in metatherians (e.g. polydolopimorphians)" is bit misleading, and so we have deleted it. As suggested, we have now added citations to Butler (2000), Butler and Hooker (2005) and recent papers by Meng and co-workers that have argued in favour of allotherian monophyly, in contrast to our results.

The reviewer correctly notes that PRSB has a tight word limit, and we feel that the recent papers by Meng et al. (2014 – "Dental and mandibular morphologies of *Arboroharamiya* (Haramiyida, Mammalia", *PLoS ONE*), Luo et al., (2015 – "Mandibular and dental characteristics of Late Triassic mammaliaform *Haramiyavia* and their ramifications for basal mammal evolution, *PNAS*), Meng and Mao (2019 - Tooth microwear and occlusal modes of euharamiyidans from the Jurassic Yanliao Biota reveal mosaic tooth evolution in Mesozoic allotherian mammals, *Palaeontology*), Wang et al. (2019 – "Cretaceous fossil reveals a new pattern in mammalian middle ear evolution", *Nature*) and Meng et al. (2020 – "A comparative study on auditory and hyoid bones of Jurassic euharamiyidans and contrasting evidence for mammalian middle ear evolution", *Journal of Anatomy*) in particular present a far more detailed and comprehensive discussion of the relevant dental and ear-jaw characters than we could provide in this paper. We have cited these papers in the text.

Lines 270-285: I agree with this discussion and highly doubt that Haramiyavia+Thomasia are sister to tritylodontids. As you note, there are many traits that differentiate Haramiyavia+Thomasia from tritylodontids. But this highlights an underlying concern – the Huttenlocker/Luo and Krause matrix characters were generated to investigate relationships of mammaliaforms, not cynodonts. (This is analogous to my above comments on therians.) There are relatively few cynodont-specific characters in those matrices. And the scarcity of sampled cynodont taxa in the matrices may help to generate the long branches and ‘attract’ Haramiyavia+Thomasia. A cynodont-focused matrix might be more likely to separate tritylodontids from Haramiyavia+Thomasia. I recommend acknowledging this concern in the text.

Authors’ response: We agree with the reviewer that the position of Haramiyavia+Thomasia as sister to tritylodontids may be due to insufficient of non-mammaliaform cynodont characters and taxa, and we have specifically stated this in our revision. However, due to page length constraints, we have moved the detailed discussion as to exactly why we consider this relationship to be unlikely to the supplementary information. The relevant section now reads:

“Our analysis places *Haramiyavia* and *Thomasia* in clade with tritylodontids, a result that may be the result of insufficient sampling of non-mammaliaform cynodont characters and taxa, and which we consider in need of further testing (see detailed discussion in supplementary information).”

Line 275: Although the chewing stroke was likely predominantly orthal in Haramiyavia, Butler (2000) suggested a significant palinal component in the chewing stroke of Thomasia.

RB: We thank the reviewer for pointing this out – we agree that the chewing stroke of *Thomasia* is likely to have been at least partly palinal (as also concluded by Mao and Meng, 2019: fig. 13A), and so we have deleted mention of this putative difference with tritylodontids (but we keep the others) – however, note that this section has now been moved to the supplementary information.

Lines 288–293: This result might also suggest that *Juramaia* might not be Jurassic in age, and instead is Early Cretaceous, but you seem hesitant to bring it up (maybe previous commenters/reviewers on this manuscript have dismissed that possibility?). Personally, I think it’s worth mentioning as a possibility, even if you doubt it’s the case. I’ve also seen *Juramaia*’s age questioned in Meng 2014 (Nat Sci Rev), and there might be other references that raise similar concerns.

Authors’ response: Reviewer 1 raised a similar point (see above), and we repeat our response here:

We realise that the reported age of *Juramaia* has been questioned in several papers, but none of these papers have presented evidence to support this conclusion besides qualitative remarks on its close morphological resemblance to the Jehol eutherians. We have quantified this resemblance for the first time by showing that, when the age of *Juramaia* is allowed to vary, our morphological clock analysis estimates the age of *Juramaia* to be similar to the Jehol eutherians. However, we emphasise that this does not show that the age of *Juramaia* is incorrect – Luo et al. (2011) clearly state that *Juramaia* is from the Daxishan/Daxigou site of Jianchang County of Liaoning Province, China, and the

age of this locality is tightly constrained as ~160 MYA. In our opinion, the only way to show that the age of *Juramaia* is incorrect is by showing that the fossil does not come from that locality, and this is not something that our paper addresses. We therefore must assume that the reported age of *Juramaia* is correct. Our analysis simply shows that the known morphology of *Juramaia* is not what we would expect given its reported age and the assumptions of our analysis (model priors etc.).

Figure 1: I recommend labeling key nodes, such as Mammaliaformes, Mammalia, Theria, and Eutheria. This will be especially important for readers to better understand which branches are being highlighted in Figure 3 (see comment below).

Authors' response: changed as requested

Figure 2: I recommend flipping the x-axis so that younger ages are to the right, especially because to stay consistent with Figure 1.

Authors' response: changed as requested

Figure 3: The results in this figure may be difficult to interpret because the branches of interest are not displayed. Consider labeling the three branches (and relevant nodes) in Figure 1, and citing Figure 1 in the Figure 3 caption.

Authors' response: changed as requested

Appendix C

The manuscript was jointly reviewed by David Grossnickle and Lucas Weaver (PhD candidate at the University of Washington).

General comments

We appreciate that the authors took the time to make major changes based on our previous comments. The paper is a significant scientific contribution, and we strongly support its publication in Proceedings B. We have no major concerns, but we list some specific suggestions below.

Specific comments

Lines 23–24: Your results certainly support the possibility that dental homoplasy was common; however, there are other traits (e.g., in the middle ear) that might be convergent as well given the tri-phyletic Allotheria result—consider rewording to something like, “Early mammal evolution may have involved multiple instances of convergent morphological evolution (e.g., in the dentition)...”.

Authors’ response: changed as requested.

Lines 30–41 and/or Discussion: We recommend you add a note about the phylogenetic analysis of Krause et al. (2020, Nature) on Adalatherium, which recovers a diphyletic Allotheria that closely resembles topology 2a from Figure 1. (Megaconus is grouped with Haramiyavia + others outside of crown Mammalia in their tree, so it’s not quite a match). We realize that Krause et al. 2020 was published after you revised your manuscript, but it’s relevant enough that we think it should be cited.

Authors’ response: We have added Krause *et al.* 2020 to the citations for topology 2b (this is the correct topology, rather than 2a).

Lines 61–63: The phrasing is somewhat misleading because it implies that haramiyidans are only known from the Late Triassic—but the haramiyidan *Thomasia* is known from the Early Jurassic and the “*eu*haramiyidans” are known from the Middle Jurassic. Consider rewording to something like, “The wide time difference between the oldest-known haramiyidans (Late Triassic) and the oldest-known multituberculates (Middle Jurassic)...”.

Authors’ response: changed as requested

Also, consider citing Butler & Hooker (2005; ref. 5) when you introduce the age of the earliest multituberculates. Averianov et al. (2020, Papers in Palaeontology) on Middle Jurassic multituberculates may also be worth citing here. And it could potentially be cited elsewhere in the paper because it presents new support for a monophyletic *eu*haramiyid + multituberculate clade (i.e. supporting Krause et al. and Wang et al. matrix results).

Authors’ response: We have cited both papers here. In addition, we also mention the Averianov paper in the discussion.

“For the Krause et al. and Wang et al. datasets, which originally recovered allotherian monophyly (figure 1, topology 1), tip dating leads to increased support for two independent lineages (figure 1, topology 2b), a topology possibly

supported by recently discovered morphological similarities between early multituberculates and euharamiyids [24]

Lines 94-95: We assume that you removed modern taxa from the Krause et al. and Wang et al. matrices. If so, it's probably worth noting the total number of characters and taxa in the pruned versions of the matrices (as you do for the Huttenlocker et al. matrix).

Authors' response: we have added the following sentence to the methods *"Extant taxa were pruned, as above, resulting in datasets of 81 taxa, 448 characters and 89 taxa, 473 characters respectively."*

Lines 144-145: You state the Wang et al. dataset resulted in a diphyletic Haramiyida, but you only talk about one of the diphyletic clades in the sentence (multituberculate + euharamiyidans). Consider including the second piece of the diphyletic clade in this sentence. Further, this would be a good point to cite Krause et al. (2020), because they recover a similar topology.

Authors' response: We have revised this sentence to:

*"with euharamiyids and multituberculates forming a clade **distant from Haramiyavia + Thomasia** (figure 1, topology 2b)"*

We cited Krause *et al.* (2020) in the introduction rather than here, along with the other studies that recovered this topology.

Lines 175-177 and Discussion: We appreciate that you followed our suggestion to test Rugosodon's predicted age after removing other early multituberculates. However, by only including a brief sentence on these results, it may not be clear to readers why you performed the supplemental analysis. We recommend you add a little more detail here or add a brief section of text on this issue in the Supplement. Also, we recommend you cite this supplemental analysis (or at least cite Fig. S12) in the relevant Discussion text (starting Line 231).

Authors' response: We have now added a new section to the supplementary information (reproduced below), giving much more detail on this analysis. This supplementary text is referenced in both the Results and Discussion, as requested.

Effect of fossil sampling on age estimates of *Juramaia* and *Rugosodon*

The result of the age estimate of *Juramaia* is striking (Fig. 3). However, it is possible that poor sampling of eutherians in the Early Cretaceous drives this result. In order to test this idea, we manufactured an equivalent situation for the multituberculate *Rugosodon* by deleting *Kuehneodon* and plagiaulacids from the dataset. This produces an approximately 40 million-year gap between *Rugosodon* from the Yanliao biota and *Sinobaatar* from the Jehol biota, equivalent to the temporal difference between *Juramaia* and *Eomaia*. We then estimated the age of *Rugosodon* using the same laplace distribution prior as used for other taxa from the Yanliao biota (main text: Material and Methods). Estimating the age of *Rugosodon* following deletion of *Kuehneodon* and plagiaulacids resulted a younger age estimate (Fig. S12), suggesting that sampling issues may indeed affect the age estimate for *Juramaia*. However, the

results are not entirely equivalent. For *Rugosodon*, the upper bound of the HPD (114.4–164.5) still overlaps with the correct age, whereas the upper bound of the HPD for *Juramaia* is much younger (106.3–137.6). Deletion of taxa results in a loss of precision for the age estimate of *Rugosodon*, representing a lack of information in the data. In contrast, the age estimation for *Juramaia* results in a distinct Early Cretaceous peak (Fig. 3), showing a strong signal in the morphological data. Nevertheless, these results show the importance of fossil sampling to constrain estimates of taxon age, and suggest the implications of the morphology of *Juramaia* may be subject to change if additional Early Cretaceous eutherian fossils are discovered.

Line 198: You are missing the closing parenthesis.

Authors' response: revised as requested

Line 202: Revise "in clade" to "in a clade"

Authors' response: revised as requested

Lines 205–222: These are great concluding remarks on the implications your study has for understanding allotherian relationships. Well done!

Authors' response: thank you!

Figure 1: Because the position of the crown Mammalia node is often referenced in the Introduction, we recommend labeling the node in each of the four phylogenies.

Authors' response: revised as requested

David Grossnickle

Lucas Weaver